# Willingness and ability to pay for healthcare insurance: A cross-sectional study of Seven Communities in East and West Africa (SevenCEWA)

Oladimeji Akeem Bolarinwa[1,2]*, Soter Ameh[2,3], Caleb Ochimana[2,4], Abayomi Olabayo Oluwasanu[2,5], Okello Samson[2,6,7], Shukri F. Mohamed[2,8], Alfa Muhihi[2,9,10], Goodarz Danaei[2,11]

1 Department of Epidemiology and Community Health, University of Ilorin, Ilorin, Nigeria, 2 Lown Scholars Program, Department of Global Health and Population, Harvard T.H. Chan School of Public Health, Boston, Massachusetts, United States of America, 3 Department of Community Medicine, College of Medical Sciences, University of Calabar, Calabar, Nigeria, 4 University Health Services, University of Ibadan, Ibadan, Nigeria, 5 Ochimana Caleb Foundation, Federal Capital Territory, Abuja, Nigeria, 6 Department of Internal Medicine, Mbarara University of Science and Technology, Mbarara, Uganda, 7 Division of Infectious Diseases and International Health, Department of Medicine, University of Virginia Health Systems, Charlottesville, Virginia, United States of America, 8 Health and Systems for Health Unit, African Population and Health Research Center (APHRC), Nairobi, Kenya, 9 Africa Academy for Public Health, Dar es Salaam, Tanzania, 10 Department of Community Health, Muhimbili University of Health and Allied Sciences, Dar es Salaam, Tanzania, 11 Department of Epidemiology, Harvard T.H. Chan School of Public Health, Boston, Massachusetts, United States of America

* bolarinwa.oa@unilorin.edu.ng

## Abstract

Willingness and ability to pay for insurance that would cover primary healthcare services has not been evaluated consistently in different African communities. We conducted a cross-sectional community health survey and examined willingness and ability to pay in 3676 adults in seven communities in four countries: Nigeria, Tanzania, Uganda and Kenya. We used an open-ended contingency valuation method to estimate willingness to pay and examined ability to pay indirectly by calculating the ratio of healthcare expenditure to total household income. Slightly more than three quarters (78.8%) of participants were willing to pay for a health insurance scheme, and just a little above half (54.7%) were willing to pay for all household members. Across sites, median amount willing to pay was $2 per person per month. A little above half (57.6%) of households in Nigeria were able to pay the premium. The main predictors of likelihood of being unwilling to pay for the health insurance scheme were increasing age [aOR 0.99 (95% CI 0.98, 1.00)], being female [0.68 (0.51, 0.92], single [0.32 (0.21, 0.49)], unemployment [0.54 (0.34, 0.85)], being enrolled in another health insurance scheme [0.45 (0.28, 0.74)] and spending more on healthcare [1.00 (0.99, 1.00)]. But being widow [2.31 (1.30, 4.10)] and those with primary and secondary education [2.23 (1.54, 3.22)] had increased likelihood of being willing to pay for health insurance scheme. Retired respondents [adjusted mean difference $-3.79 (-7.56, -0.02)], those with primary or secondary education [$-3.05 (-5.42, -0.68)] and those with high healthcare expenditure [$0.02 (0.00, 0.04)] predicted amount willing to pay for health insurance scheme. The

**Data Availability Statement:** https://dataverse.
harvard.edu/dataset.xhtml?persistentId=doi:10.
7910/DVN/MLE2CR.

**Funding:** This study was funded by the Bernard
Lown Scholars in Cardiovascular Health Program
at the Department of Global Health and Population,
Harvard T.H. Chan School of Public Health, Boston,
MA, USA with award number BLSCHP-1707. The
Funder has no other role in the conduct of this
study other than the fund provided.

**Competing interests:** The authors have declared
that no competing interests exist.

willingness to pay for health insurance scheme is high among the seven communities studied in East and West Africa with socio-demography, economic and healthcare cost as main predictive factors.

## Introduction

The Sustainable Development Goals (SDGs) tasks the governments and the international community to attain the Universal Health Coverage (UHC) under its target 3.8 [1]. In spite of this, achieving UHC in Africa requires Governments shifting from a predominantly Out of Pocket (OoP) healthcare finance to a prepayment plan [2–5]. Despite the global push and various regional and national efforts, the coverage of health insurance is still low among many African countries [2]. For African countries to increase the reliance on prepayment and pooling scheme to finance the healthcare costs, it is mandatory that the member countries should spend at least $86 per capital and a minimum of 5% of the Gross Domestic Products (GDP) on healthcare [6, 7].

A recent global review of the funding and services required to achieve UHC estimated that sub-Saharan Africa requires over $120 billion additional investment to meet the UHC goal [8]. For instance, Nigeria will need $24.7 billion (2.2% of GDP), Kenya $5.6 billion (3.7%), Tanzania $1.9 billion (1.2%), and Uganda $4.3 billion (5.1%) [8]. Whereas only a few countries in the region have made strides in achieving this level of investment and many are still unable to meet the Abuja Declaration target of over 15% of National budget allocation to health set by the African Governments. In addition, most African countries including Nigeria, Kenya, Uganda and Tanzania are struggling to provide health insurance for and as well lagging for their population. In many of these countries, less than 10% of the population have health insurance coverage [2, 9]. This invariably creates inequalities to healthcare access among the population.

For a successful social insurance scheme, tax revenue generation is cardinal. African countries do not have a large tax-paying population thereby making health insurance as a social schemes more difficult to establish. Therefore, if the continent has to achieve UHC, it is reasonable to consider alternative models where users bear some of the healthcare cost, for example by paying a premium to community-based health insurance schemes.

The current reality is that some African countries that have steered policy directions towards prepayment and insurance, with the view to achieve UHC are still battling with the best methods to mobilize and engage their people. Another spectrum of challenge is that Health Insurance schemes in African countries and other low-income countries are frequently commenced without strong empirical information that could help benchmark cost-sharing policies. In most cases, potential and future enrollees are rarely engaged before the commencement of the schemes resulting in poor acceptability [10, 11]. These can be attributed amongst other reasons to inadequate knowledge regarding participants' willingness and ability to pay for a prepayment plan.

To ensure the sustainability of the Health Insurance policy among African states, countries have been encouraged to strategize and plan for the scheme according to economic, sociocultural and political situations of the target population as well as their perceptions and preferences regarding quality of care [12, 13]. This will consequently increase the acceptability of the prepayment or health insurance scheme.

A key factor that determines the sustainability of a health insurance scheme is the amount of the premium, that is, the Willingness to Pay (WTP) for such service [14]. Similarly, it is essential to measure the Ability to Pay (ATP) within potential participants in the scheme

without bearing a heavy financial burden [11, 14, 15]. In fact, there is evidence that participants may not pay what they claimed they were willing to pay. In SSA, evidence on WTP for health insurance is scarce and previous studies have employed different methodologies therefore reporting varying and sometimes inconsistent results across the regions. For instance, evidence from East and West African countries reported varying values of WTP from 58% in Uganda to 89% in Nigeria [16–19].

In the literature, the monthly amount willing to pay out for health insurance scheme ranges from $0.4 in Tanzania [20], $1.56 in Uganda [17] and $1.68 in Nigeria [21]. Varying factors were reported as determinants of willingness to pay for insurance scheme. Gender, education, age, occupation, place of residence and household income were recurring factors across the regions. The available studies provided little in-depth information on consumers' willingness and ability to pay for the health insurance scheme [22] and the studies that reported ATP were invariably scarce [21]. Therefore, we conducted a multi-site cross-sectional survey in seven communities across four African countries to assess WTP and ATP for insurance scheme and we also described important factors that determine the reported WTP. With Africa countries making big stride to increase health insurance scheme coverage across the continent, the findings of our study will guide health policy makers, insurance companies, and local and national governments across Africa on premium policies and to identify the populations that will require subsidies.

## Materials and methods

The data for the study was obtained from the household survey in 7 communities in Kenya, Nigeria, Tanzania and Uganda. The four sites from Nigeria were; Ikire town, a peri-urban community in Osun state in South-west; Olorunda Abba, a rural community in Oyo state in South-west; Ikpok Ikpa, a rural community in Cross river state in South-south and Ogane Uge community, a rural community in Kogi state in North-central. In Kenya, the site was Viwandani community, an urban slum in Nairobi, in Uganda it was Soroti community from Soroti municipality in the east of the country and in Tanzania, it was Ukonga, a semi-urban community from Ilala municipal, Dar es Salaam region. Details for each community can be seen in Table 2 of another manuscript [23].

Participants were eligible if they were above 18 years and permanent residents of the study communities. Full institutional review board approval was obtained from University of Ilorin Teaching Hospital (UITH) Ethical Research Committee (ERC) with approval no ERC/PAN/2018/03/1787. An approved written informed consent was obtained from all the respondents in the presence of witnesses. A total of 3,676 respondents chosen from the households were interviewed from all the study sites, 2,136 of whom were from the 4 study sites in Nigeria, 300 from Kenya, 453 from Tanzania and 787 from Uganda. Each site estimated the sample size using standard techniques. The study respondents were obtained from the study populations using random sampling techniques. Pretested questionnaires were used across the study sites and were validated, adapted and translated to local language. Open ended contingency valuation method was used to estimate WTP by the respondent while ATP healthcare was indirectly estimated from ratio of the healthcare expenditure to household income. A respondent was deemed unable to pay for healthcare services when this ratio is greater than 5% [15]. To examine willingness to participate a direct question was asked; *"Would you be willing to participate in the health insurance scheme and pay a health insurance premium for you or other members of your household?"* Similarly, in all study sites except for the one in Kenya, WTP was asked using a direct question; *"What is the maximum amount that you are certain that you would be willing to pay for the health insurance premium, for each person of your household member for one*

*month*?" In Kenya, the 'payment ladder method' was used. The respondents were provided with a set of bid amounts and they were asked to choose the amount that they would be willing to pay. All cost related questions were asked in local currencies and converted to $ using prevailing exchange rate. Data was analyzed using SPSS software.

For descriptive analysis, both disaggregated (by study sites) and combined socio-economic, health facility utilization and willingness to pay results were presented. To analyze the explanatory factors for the willingness to pay and the amount willing to pay (in $), Heckman's two-stage model was adopted for the analysis [24]. In step one, hierarchical multivariate logic regression was used to model the predictors of willingness to pay for health insurance scheme among the communities while controlling for age and household income as covariates. The outcome variables is a dichotomous variable of "willing to pay for health insurance scheme (1) as against not willing to pay for health insurance scheme (0). Dummy variables were created for independent categorical variable. In the second step, we examined the predictors of the amount households are willing to pay (in $) using hierarchical multivariate linear regression while controlling for age (as covariate). Only households that are willing to pay in the stage one model entered into the stage two. In the regression analysis, assumptions of independence of variables, linearity (using correlation coefficients) and normality was ascertained. In addition, multicollinearity and interactions were checked using tolerance of close to 1 or greater than $(1-R^2)$ in the model. Parsimonious variables were fed into the model using literature evidence, researchers' experiences and correlated variables.

## Results

Mean (SD) age of respondents was 40 (±16) years across the sites, females constituted 60.6% of the sample while 70.9% of the respondents were either married or living with a partner (Table 1). Only 14.6% of the respondents had no formal education and about a quarter had university education. Across all sites, more than half (59.7%) of participants were self-employed, compared with only a quarter in the urban slum and semi-urban sites in Kenya and Uganda. About a fifth (19.5%) were unemployed with the highest unemployment (49.9%) reported among Uganda respondents. Average household income (across Nigerian sites) was $78 a month. Two-thirds (77.5%) of participants owned a mobile phone, a third (33.9%) had a bank account and internet use was 23%.

More than two-thirds (70.1%) of the participants across sites perceived themselves to be in good or excellent health (Table 2). Over 60% of the households used public hospitals and clinics for healthcare services and 13.8% were dissatisfied with the overall experience in the healthcare facilities. Average household expenditure in the last 3 months was $10 per month with the highest expenditure ($18) reported in Ogane-Uge and lowest ($7) in Ikire and Ikpok Ikpa, both in Nigeria. On the average, only 10.6% were ever enrolled in a health insurance scheme with the highest enrollment rate in Viwandani, Kenya at 43%.

Slightly more than three quarters (78.8%) of participants were willing to participate in a health insurance scheme, and 54.7% were willing to pay for all household members. Across sites, median amount willing to pay for health insurance by the household was $2 per person per month with highest ($7) reported in Ogane-Uge (Table 2). A little above half (57.6%) of the households from Nigerian sites had ability to pay for the health insurance. This varied across the four sites in Nigeria with Ikpok Ikpa having the highest ability to pay (82.2%).

Female respondents were less willing (77.1%) to pay for health insurance scheme than males (81.3%) (Table 3). Self-employed respondents were most likely to be willing to pay for the scheme (80.6%) compared with other occupational groups. They were also willing to pay more ($5.50 per person per month) than those in the government ($3.11) and private ($3.40)

**Table 1. Socio-economic characteristics of the households and participants in Seven Communities in East and West Africa (SevenCEWA), 2018.**

| Socio-demography/ Study site | Kenya Viwandani (Urban Slum) (n = 300) | Nigeria Ikire (Semi-urban) (n = 502) | Olorunda-Abaa (Rural) (n = 732) | Ogane-Uge (Rural) (n = 410) | Ikpok Ikpa (Rural) (n = 492) | Tanzania Ukonga (Semi-urban) (n = 453) | Uganda Soroti (Semi-urban) (n = 787) | Total N = 3676 |
|---|---|---|---|---|---|---|---|---|
| Age (mean in years ± SD) | 35 ±11 | 48 ±18 | 41 ±13 | 39 ±20 | 39 ±15 | 44 ±14 | 34 ±12 | 40 ±16 |
| Female (%) | 145 (48.3) | 278 (55.4) | 469 (64.1) | 212 (51.7) | 240 (48.8) | 325 (71.7) | 557 (70.8) | 2226 (60.6) |
| Married or living with a partner (%) | 171 (57.0) | 370 (61.4) | 657 (89.8) | 302 (73.7) | 302 (61.4) | 296 (65.3) | 509 (64.7) | 2607 (70.9) |
| No education (%) | 4 (1.3) | 105 (20.9) | 48 (6.6) | 107 (26.1) | 113 (23.0) | 57 (12.6) | 101 (12.8) | 535 (14.6) |
| Secondary school (%) | 156 (52.0) | 196 (39.0) | 310 (42.3) | 125 (30.5) | 121 (24.6) | 30 (6.6) | 237 (30.1) | 1175 (32.5) |
| College or university (%) | 20 (6.7) | 71 (14.1) | 216 (29.5) | 22 (5.4) | 47 (9.6) | 245 (54.1) | 244 (31.0) | 865 (23.5) |
| Self-employed (%) | 73 (24.3) | 403 (80.3) | 579 (79.1) | 312 (76.1) | 322 (65.4) | 275 (60.7) | 230 (29.2) | 2194 (59.7) |
| Unemployed (%) | 37 (12.3) | 18 (3.6) | 39 (5.3) | 66 (16.1) | 35 (7.1) | 129 (28.5) | 393 (49.9) | 717 (19.5) |
| Household monthly income in $ (median (IQR)) | NA | 65 (134) | 67 (131) | 23 (70) | 98 (137) | NA | NA | 78 (130) |
| Mobile phone ownership (%) | 273 (91.0) | 450 (89.6) | 683 (93.4) | 199 (48.5) | 281 (57.1)) | NA | NA | 1886 (77.5) |
| Operating account in financial institution (%) | 137 (45.7) | 192 (38.2) | 349 (47.7) | 58 (14.1 | 89 (18.1) | NA | NA | 825 (33.9) |
| Access to internet (%) | 151 (50.3) | 60 (12.0) | 246 (33.7) | 55 (13.4) | 47 (9.6) | NA | NA | 559 (23.0) |

**NA**: Data not available.

employments; and more than those retired ($3.06). Respondents with primary/secondary or tertiary education were more likely to be willing to pay for the scheme (almost 80%) than those without formal education (74%). However, among those willing to pay, respondents without formal education were willing to pay more ($6.75) compared with those with formal education.

In multivariate analyses of willingness to pay after adjusting for age and household income, the likelihood of being willing to pay for the health insurance scheme decreased with increasing age [aOR 0.99 for each year of age (95% CI 0.98, 1.00)], being female [0.68 (0.51, 0.92], single compared with married [0.32 (0.21, 0.49)], unemployed compared with self-employed [0.54 (0.34, 0.85)], being enrolled in another health insurance scheme [0.45 (0.28, 0.74)] and spending more on healthcare [1.00 per $ more spent (0.99, 1.00)] (Table 4). In contrast, widows [2.31 (1.30, 4.10)] and those with primary and secondary education [2.23 (1.54, 3.22)] had increased likelihood of being willing to pay for health insurance scheme.

In multivariate analyses of those who said they were willing to pay, mean amount willing to pay was lower in retired respondents compared with self-employed [adjusted mean difference $-3.79 per person per month (-7.56, -0.02)] and in those with primary or secondary education compared with those with no formal education [$-3.05 (-5.42, -0.68)]. Conversely, those with high healthcare expenditure [$0.02 per $ more spent on healthcare per month (0.00, 0.04)] were willing to pay higher amounts for health insurance scheme (Table 5).

**Table 2. Healthcare utilization, enrolment in health insurance and willingness to pay for insurance in Seven Communities in East and West Africa (SevenCEWA), 2018.**

| Healthcare utilization/Health insurance enrolment | Kenya | Nigeria | | | | Tanzania | Uganda | Total |
|---|---|---|---|---|---|---|---|---|
| | Viwandani | Ikire | Olorunda Abaa | Ogane-Uge | Ikpok Ikpa | Ukonga | Soroti | |
| Perceived good/excellent health | 240 (80.0) | 412 (82.1) | 510 (69.7) | 280 (68.3) | 421 (85.6) | 303 (66.9) | 410 (52.1) | 2575 (70.1) |
| Utilization of private hospital/clinics (%) | 94 (31.3) | 185 (38.2) | 160 (21.9) | 39 (11.3) | 10 (2.0) | 241 (53.2) | 75 (9.6) | 804 (22.4) |
| Utilization of public hospital/clinics (%) | 193 (64.3) | 223 (46.1) | 466 (63.7) | 305 (88.7) | 294 (59.8) | 155 (34.2) | 644 (82.0) | 2280 (63.5) |
| Not satisfied with overall experience in the health facilities (%) | 49 (19.1) | 8 (1.7) | 319 (43.6) | 23 (7.0) | 5 (1.0) | 13 (2.9) | 69 (13.8) | 486 (13.8) |
| Household health expenditure (US$ per month) (median (IQR)) | 11 (19) | 7 (20) | 8 (18) | 18 (38) | 7 (13) | 11 (18) | 13 (23) | 10 (21) |
| Enrollment in health Insurance (%) | 129 (43.0) | 8 (1.6) | 57 (13.7) | 40 (10.4) | 10 (2.0) | 79 (17.4) | 29 (3.7) | 352 (10.6) |
| Willingness to pay for health insurance (%) | 292 (97.3) | 480 (95.6) | 476 (65.0) | 268 (65.4) | 428 (87.0) | 424 (93.6) | 528 (67.1) | 2896 (78.8) |
| Willing to pay for all household members (%) | NA | 332 (70.0) | 88 (12.0) | 209 (64.1) | 384 (78.0) | 201 (45.9) | 560 (71.2) | 1774 (54.7) |
| Amount willing to pay US$ per person per month (median (IQR)) | 2 (2) | 2 (2) | 2 (2) | 7 (10) | 2 (2) | 1 (2) | 1 (5) | 2 (2) |
| Ability to pay for Insurance Scheme (%)* | NA | 108 (53.7) | 221 (57.7) | 23 (16.3) | 221 (82.2) | NA | NA | 573 (57.6) |

**NA**: Data not available.

## Discussion

In this multi-site community-level study of health and healthcare utilization, we found that a large majority of households, that were mostly self-employed and educated, were willing to pay for the proposed health insurance scheme and just a little above half of them were willing to pay for other members of the household. The high willingness to pay is similar to other studies from Kenya, Uganda [17], Tanzania [16] and Nigeria [19, 21]. The amount willing to be paid (payout) across the 7 sites was $2 per person per month. This is similar to $1.68 reported in a prior Nigerian study [21] but much more than the $1.56 [17] annual payout reported from Uganda [17] and $0.4 from Tanzania [20]; and slightly higher than $1.5 monthly payout from a voluntary scheme in Ethiopia [25]. Most premiums established for health insurance schemes in sub-Saharan Africa were not based on empirical evidence or market surveys [26]. Across Nigerian sites, average household health expenditure was $10 per month and 57% of households reflected ability to pay the proposed health insurance premiums. This proportion is consistent with 65% ability to pay reported from a previous study in Nigeria [21]. Conversely, our finding implied that almost half (43%) of the household are currently unable to sustain healthcare expenditures and are more likely to suffer from catastrophic health expenditure [15].

The major strength of applying ability to pay to make decision on the willingness to pay for health services or commodities lies majorly in the economic and policy decisions. This will give an indication on the available fiscal space for the health insurance scheme funding [21]. In this instance, the health demands and purchasing power of the consumers are estimated as against the earning of a health consumer. This invariably reflect the burden of the health expenditure on the earning and could be used to make market and policy decisions on health insurance packages and benefits. However, an obvious limitation lies in the diverse nature of household earning data which could pose impractical to being correctly estimated [15].

**Table 3. Factors associated with Willingness to pay and amount willing to pay ($) among respondents in Seven Communities in East and West Africa (Seven-CEWA), 2018.**

| Factors | Willingness to Pay for health Insurance | | | | Amount willing to pay for health insurance | | |
|---|---|---|---|---|---|---|---|
| | Not Willing n (%) | Willing n (%) | $X^2$ /t-test | p-value | Mean (S.D) /correlation | t-test/F-test | p-value |
| Age (years) mean (S.D) | 39.78 (15.01) | 40.34 (16.97) | -0.848 | 0.363 | -0.07 | - | 0.692 |
| *Gender* | | | | | | | |
| Male | 271 (18.7) | 1179 (81.3) | 9.162 | 0.002 | 5.59 (16.40) | 2.386 | 0.017 |
| Female | 509 (22.9) | 1717 (77.1) | | | 4.24 (14.49) | | |
| *Marital Status* | | | | | | | |
| Married | 550 (21.1) | 2056 (78.9) | 3.072 | 0.381 | 5.11 (16.24) | 2.286 | 0.077 |
| Divorce | 46 (20.9) | 174 (79.1) | | | 3.22 (12.55) | | |
| Widow | 64 (18.7) | 278 (81.3) | | | 3.05 (4.92) | | |
| Single | 120 (23.6) | 388 (76.4) | | | 5.01 (11.69) | | |
| *Employment* | | | | | | | |
| Self employed | 425 (19.4) | 1769 (80.6) | 76.948 | <0.001 | 5.50 (17.66)* | 3.253 | 0.011 |
| Government | 48 (11.0) | 387 (39.0) | | | 3.11 (5.05)* | | |
| Private | 41 (25.0) | 123 (75.0) | | | 3.40 (4.73)* | | |
| Unemployment | 223 (31.1) | 494 (68.9) | | | 4.36 (10.85) | | |
| Retired | 43 (25.9) | 123 (74.1) | | | 3.06 (6.18)* | | |
| *Education* | | | | | | | |
| No education | 138 (25.8) | 397 (74.2) | 8.299 | 0.016 | 6.75 (25.77)* | 5.099 | 0.006 |
| Primary/Secondary | 460 (20.8) | 1756 (79.2) | | | 4.25 (11.06)* | | |
| Tertiary | 182 (19.7) | 743 (80.3) | | | 4.97 (13.73) | | |
| *General Health Perception* | | | | | | | |
| Good | 532 (20.7) | 2044 (79.3) | 1,653 | 0.199 | 4.86 (14.51) | 0.386 | 0.699 |
| Poor | 248 (22.5) | 852 (77.5) | | | 4.63 (15.38) | | |
| *Satisfaction with healthcare* | | | | | | | |
| Not satisfied | 143 (29.4) | 343 (70.6) | 22.555 | <0.001 | 5.45 (15.02) | | |
| Satisfied | 637 (20.0) | 2553 (80.0) | | | 4.71 (14.73) | 0.873 | 0.383 |
| *Enrollment health insurance* | | | | | | | |
| Enrolled | 41 (11.6) | 311 (88.4) | 22.044 | <0.001 | 4.59 (8.50) | -0.041 | 0.968 |
| Not enrolled | 512 (17.3) | 2455 (82.7) | | | 4.62 (14.14) | | |
| Household income ($) | 96.89 (2.39) | 106.81 (2.29) | -0.823 | 0.411 | 0.01 | - | 0.929 |
| Healthcare expenditure cost ($) | 17.36 (53.69) | 17.96 (45.69) | -0.286 | 0.775 | 0.03 | - | 0.106 |

* Significant combination at post hoc.

Another limitation is that ability to pay estimate do not give adequate information on the opportunity cost incurred by the households. It is believed to have hidden information on the forgone alternatives to resources expended on healthcare cost [15].

The factors determining willingness to pay and amount willing to pay by the households members were discussed under four characteristics. Firstly, age, gender and marital status were found to be main socio-demographic factors in this study. Increasing age, being female and single reduced the likelihood of being willingness to pay for health insurance scheme. Our findings are consistent with studies in Africa which reported that increasing age reduces the likelihood of community members to accept to join or pay for health insurance scheme [20, 27]. While some studies reported reduced likelihood of female to join or pay for health insurance scheme like we reported [20, 22], others reported contrary [27–29]. However, widowed participants were much more likely to be willing to pay for the scheme.

**Table 4. Predictors of willingness to pay for health insurance scheme in Seven Communities in East and West Africa (SevenCEWA), 2018, (N = 1745).**

| Predictors | Adjusted Odd Ratio | 95% CI | | p-value |
|---|---|---|---|---|
| Age (year) | 0.991 | 0.981 | 1.000 | 0.052 |
| *Male (Ref)* | | | | |
| Female | 0.684 | 0.509 | 0.920 | 0.012 |
| *Marital status* | | | | |
| Married *(Ref)* | | | | |
| Divorced | 1.272 | 0.609 | 2.657 | 0.522 |
| Widowed | 2.313 | 1.304 | 4.101 | 0.004 |
| Single | 0.323 | 0.212 | 0.493 | <0.001 |
| *Employment* | | | | |
| Self-employed *(Ref)* | | | | |
| Government | 1.206 | 0.612 | 2.374 | 0.589 |
| Private | 1.117 | 0.525 | 2.378 | 0.773 |
| Unemployed | 0.537 | 0.340 | 0.850 | 0.008 |
| Retired | 0.728 | 0.413 | 1.282 | 0.272 |
| *Education* | | | | |
| No education *(Ref)* | | | | |
| Primary /Secondary education | 2.229 | 1.543 | 3.222 | <0.001 |
| College/University | 1.538 | 0.956 | 2.473 | 0.076 |
| Utilize other health facility *(Ref)* | | | | |
| Utilize Public facility | 1.099 | 0.793 | 1.524 | 0.570 |
| Not satisfied with healthcare (Ref) | | | | |
| Satisfied with healthcare | 0.722 | 0.435 | 1.201 | 0.210 |
| Good health *(Ref)* | | | | |
| Poor health | 1.335 | 0.943 | 1.888 | 0.103 |
| Not enroll in insurance *(Ref)* | | | | |
| Enroll in insurance | 0.453 | 0.279 | 0.737 | 0.001 |
| Cost of total health expenditure ($) | 0.996 | 0.994 | 0.998 | 0.001 |
| Household income ($) | 1.000 | 0.999 | 1.000 | 0.574 |

Employment status and education were important socio-economic factors. While unemployed household members were half as likely to pay for the scheme, those with primary or secondary education were twice more likely to pay compared with those without formal education. Expectedly, higher education and employment have been shown to increase the likelihood of accepting health insurance scheme in the region [20, 22, 27, 29]. However, we found that having primary or secondary education reduced the amount respondents were willing to pay for the scheme. Similarly, retire members were willing to pay less for the scheme. We found that respondents with previous enrolment in health insurance were less likely to be willing to pay for health insurance scheme. This unexpected finding differ from other studies [22] that reported increased likelihood to pay for the scheme among those with experience with health insurance scheme. Finally, perceived healthcare need measured by healthcare expenditure was found to be associated with both willingness to pay and the amount willing to pay for the scheme. Though the higher healthcare expenditure will likely reduce the willingness of the household members to pay for the scheme, however those with higher healthcare expenditure are willing to pay higher amount. This is consistent with other studies [21, 22] that reported that perceived needs by the community members can greatly influence their decision to a higher amount for the health insurance scheme.

**Table 5. Predictor of amount willing to pay (in $) for health insurance scheme among those willing to pay in Seven Communities in East and West Africa (Seven-CEWA), 2018, (N = 1744).**

| Predictors | Mean difference ($ per person per month) | 95% CI | | p-value |
|---|---|---|---|---|
| Age (year) | 0.002 | -0.061 | 0.064 | 0.962 |
| Male *(Ref)* | | | | |
| Female | 0.156 | -1.645 | 1.958 | 0.865 |
| *Marital status* | | | | |
| Married *(Ref)* | | | | |
| Divorced | -0.997 | -5.066 | 3.073 | 0.631 |
| Widowed | -2.418 | -5.729 | 0.893 | 0.152 |
| Single | 1.536 | -1.589 | 4.664 | 0.335 |
| *Employment* | | | | |
| Self-employed *(Ref)* | | | | |
| Government | -2.770 | -6.671 | 1.130 | 0.164 |
| Private | -3.515 | -7.854 | 0.825 | 0.112 |
| Unemployed | -0.894 | -4.306 | 2.518 | 0.607 |
| Retired | -3.790 | -7.562 | -0.017 | 0.049 |
| *Education* | | | | |
| No education *(Ref)* | | | | |
| Primary education | -3.047 | -5.418 | -0.676 | 0.012 |
| College/University | -1.579 | -4.780 | 1.623 | 0.334 |
| Utilize other health facility *(Ref)* | | | | |
| Utilize Public facility | -0.159 | -2.127 | 1.808 | 0.874 |
| Not satisfied with healthcare *(Ref)* | | | | |
| Satisfied with healthcare | 0.234 | -2.328 | 2.796 | 0.858 |
| Good health *(Ref)* | | | | |
| Poor health | 0.621 | -1.462 | 2.704 | 0.559 |
| Household income ($) | 0.000 | -0.003 | 0.004 | 0.870 |
| Cost of total health expenditure ($) | 0.022 | 0.000 | 0.043 | 0.048 |

Our results indicate that although willingness to pay and the amount willing to pay among low-income populations in these seven communities is fairly high, households with elderly or female heads and those who are unemployed may not be willing or able to pay even a small premium for health insurance. The effects of socio-demography and socio-economic factors on willingness to pay for insurance scheme have been documented in African studies [17, 22, 27–30] but the direction of the influence has been mixed. This calls for subsidy, co-payment or social health insurance scheme from the local or national governments. As the government in some countries in the sub-region are expanding and enacting policies to engage the informal sector in the health insurance scheme [31, 32] these findings will be useful for implementation and public mobilization. We also observed that participants who spent more on healthcare were willing to pay more for a health insurance scheme. Indirectly, this inferred that the cost expended on care look more sensitive to amount willing to pay than episodes of illness (acute and chronic illness). The willingness to pay and the amount willing to pay should be taken into account to guide health policy makers, insurance companies, and local and national governments on premium policies and to identify the populations that will require subsidies.

## Conclusion

In this cross-sectional and comprehensive survey of seven diverse communities in four countries from East and West Africa, we found a substantial proportion of the population willing to

pay for a health insurance scheme with a median willingness to pay of $2 per person per month. Socio-demographic and economic factors, as well as perceived health needs and prior experience with health insurance schemes affected respondents' willingness to pay and amount they were willing to pay as premium. Providing government subsidies or establishing a social health insurance scheme should be considered as alternative scenarios to tax-based health insurance in achieving the Universal Health Coverage goal in the region.

## Author Contributions

**Conceptualization:** Oladimeji Akeem Bolarinwa, Caleb Ochimana, Okello Samson, Alfa Muhihi, Goodarz Danaei.

**Formal analysis:** Oladimeji Akeem Bolarinwa, Goodarz Danaei.

**Investigation:** Soter Ameh, Caleb Ochimana, Abayomi Olabayo Oluwasanu, Okello Samson, Shukri F. Mohamed, Alfa Muhihi.

**Methodology:** Oladimeji Akeem Bolarinwa, Soter Ameh, Caleb Ochimana, Abayomi Olabayo Oluwasanu, Okello Samson, Shukri F. Mohamed, Alfa Muhihi, Goodarz Danaei.

**Project administration:** Oladimeji Akeem Bolarinwa, Soter Ameh, Caleb Ochimana, Abayomi Olabayo Oluwasanu, Okello Samson, Shukri F. Mohamed, Alfa Muhihi, Goodarz Danaei.

**Supervision:** Goodarz Danaei.

**Writing – original draft:** Oladimeji Akeem Bolarinwa.

**Writing – review & editing:** Oladimeji Akeem Bolarinwa, Soter Ameh, Caleb Ochimana, Abayomi Olabayo Oluwasanu, Okello Samson, Shukri F. Mohamed, Goodarz Danaei.

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
