## [Decision Letter · Decision Letter 0]

31 Aug 2021

 PGPH-D-21-00244 Willingness and ability to pay for healthcare insurance: a cross-sectional study of Seven Communities in East and West Africa (SevenCEWA) PLOS Global Public Health

Dear Dr. Bolarinwa,

Thank you for submitting your manuscript to PLOS Global Public Health. After careful consideration, we feel that it has merit but does not fully meet PLOS Global Public Health’s publication criteria as it currently stands. Therefore, we invite you to submit a revised version of the manuscript that addresses the points raised during the review process.

We look forward to receiving your revised manuscript.

Kind regards,

Paolo Angelo Cortesi, PhD

Academic Editor

Journal Requirements:

Additional Editor Comments (if provided):

Reviewers' comments:

Reviewer's Responses to Questions

**Comments to the Author**

1. Does this manuscript meet PLOS Global Public Health’s publication criteria? Is the manuscript technically sound, and do the data support the conclusions? The manuscript must describe methodologically and ethically rigorous research with conclusions that are appropriately drawn based on the data presented.

Reviewer #1: Yes

Reviewer #2: Yes

2. Has the statistical analysis been performed appropriately and rigorously?

Reviewer #1: Yes

Reviewer #2: Yes

3. Have the authors made all data underlying the findings in their manuscript fully available (please refer to the Data Availability Statement at the start of the manuscript PDF file)?

Reviewer #1: Yes

Reviewer #2: Yes

4. Is the manuscript presented in an intelligible fashion and written in standard English?

Reviewer #1: Yes

Reviewer #2: Yes

5. Review Comments to the Author

Reviewer #1: Overall, it is a good study and manuscript.

However, contingency valuation method used in measuring WTP in the study was not clearly explained in the methodology section and not presented in the result section.

Reviewer #2: Yes. However, some minor revisions are required. Author’s attention has been drawn to sentences that need to be reviewed. E.g.:

Line 68 to 71 “Retired respondents …………..health insurance scheme.”

Line 86 to 89 “Whereas only a few ……..African Governments.”

Line 89 to 90 “In addition, …………… lagging for their population.”

Line 93 “For a success….. generation is cardinal” (‘successful’ for ‘success’)

Line 118 to 120

Line 151 “………while ATP for healthcare………”

Line 257 to 260 “This is similar to …………scheme in Ethiopia.”

Line 274 “……pose impractical challenges to being ……”

Line 334 “……securely stored in in both hard….”

This is an interesting paper for LMICs working at establishing some form of health insurance to attain UHC. However, payment of premiums is not the only way to go. I find that the paper does not come out strongly on equity and social justice, that which UHC and SDG 3 partly aims at. The crux of the paper appears to be “… if people are willing to pay and have the ability to pay some amount as premium, they can enrol on a social health insurance scheme towards the attainment of UHC and SDG3”. “And so, it will be good that policy makers consider this.” [If this is not what the authors are saying, then it is not very clear to me]. In practice, it does not work that way (policy implementation is not that linear). What happens to those who cannot afford these premiums? What are the reasons why some people are not willing to pay or enrol onto insurance schemes as found in the results? I find the discussion of the findings (mainly) and introduction, a bit deficient in this regard.

6. PLOS authors have the option to publish the peer review history of their article (what does this mean?). If published, this will include your full peer review and any attached files.

**Do you want your identity to be public for this peer review?** For information about this choice, including consent withdrawal, please see our Privacy Policy.

Reviewer #1: No

Reviewer #2: **Yes: **Abigail Nyarko Codjoe Derkyi-Kwarteng

---

## [Decision Letter · Decision Letter 1]

19 Oct 2021

Willingness and ability to pay for healthcare insurance: a cross-sectional study of Seven Communities in East and West Africa (SevenCEWA)

PGPH-D-21-00244R1

Dear Dr. Bolarinwa,

We're pleased to inform you that your manuscript has been judged scientifically suitable for publication and will be formally accepted for publication once it meets all outstanding technical requirements.

Within one week, you'll receive an e-mail detailing the required amendments. When these have been addressed, you'll receive a formal acceptance letter and your manuscript will be scheduled for publication.

An invoice for payment will follow shortly after the formal acceptance. To ensure an efficient process, please log into Editorial Manager at https://www.editorialmanager.com/pgph/ click the 'Update My Information' link at the top of the page, and double check that your user information is up-to-date. If you have any billing related questions, please contact our Author Billing department directly at authorbilling@plos.org.

Kind regards,

Paolo Angelo Cortesi, PhD

Academic Editor

Additional Editor Comments (optional):

Reviewers' comments:

Reviewer's Responses to Questions

**Comments to the Author**

1. If the authors have adequately addressed your comments raised in a previous round of review and you feel that this manuscript is now acceptable for publication, you may indicate that here to bypass the “Comments to the Author” section, enter your conflict of interest statement in the “Confidential to Editor” section, and submit your "Accept" recommendation.

Reviewer #1: All comments have been addressed

Reviewer #2: All comments have been addressed

2. Does this manuscript meet PLOS Global Public Health’s publication criteria? Is the manuscript technically sound, and do the data support the conclusions? The manuscript must describe methodologically and ethically rigorous research with conclusions that are appropriately drawn based on the data presented.

Reviewer #1: Yes

Reviewer #2: Yes

3. Has the statistical analysis been performed appropriately and rigorously?

Reviewer #1: Yes

Reviewer #2: Yes

4. Have the authors made all data underlying the findings in their manuscript fully available (please refer to the Data Availability Statement at the start of the manuscript PDF file)?

Reviewer #1: Yes

Reviewer #2: Yes

5. Is the manuscript presented in an intelligible fashion and written in standard English?

Reviewer #1: Yes

Reviewer #2: Yes

6. Review Comments to the Author

Reviewer #1: (No Response)

Reviewer #2: This is an interesting paper. Well done with addressing the reviewer comments. However, I think it would be great if authors read a bit more about health insurance financing in LMICs to appreciate issues on equity and social justice in relation to financing options better. Actually, financing from general taxes, exemptions and subsidies are good for vulnerable groups, while co-pays and payment of premiums, depending on context, are not too good for vulnerable groups. Reading on some more studies from Tanzania, Kenya, Ghana and maybe South Africa, might help (if you wish) to incorporate the equity sentences better into the paper.

7. PLOS authors have the option to publish the peer review history of their article (what does this mean?). If published, this will include your full peer review and any attached files.

**Do you want your identity to be public for this peer review?** For information about this choice, including consent withdrawal, please see our Privacy Policy.

Reviewer #1: No

Reviewer #2: No
